

# Comparative analysis of magnetic induction based communication techniques for wireless underground sensor networks

Pratap S. Malik[1], Mohamed Abouhawwash[2,3], Abdulwahab Almutairi[4], Rishi Pal Singh[1] and Yudhvir Singh[5]

[1] Department of Computer Science & Engineering, Guru Jambheshwar University of Science & Technology, Hisar, Haryana, India
[2] Department of Computational Mathematics, Science, and Engineering (CMSE), Michigan State University, East Lansing, MI, United States
[3] Department of Mathematics, Faculty of Science, Mansoura University, Mansoura, Egypt
[4] School of Mathematics, Unaizah College of Sciences and Arts, Qassim University, Buraydah, Al-Qassim, Saudi Arabia
[5] Department of Computer Science & Engineering, Maharshi Dayanand University, Rohtak, Haryana, India

Corresponding author
Pratap S. Malik,
pratap.malik@gjust.org

## ABSTRACT

A large range of applications have been identified based upon the communication of underground sensors deeply buried in the soil. The classical electromagnetic wave (EM) approach, which works well for terrestrial communication in air medium, when applied for this underground communication, suffers from significant challenges attributing to signal absorption by rocks, soil, or water contents, highly varying channel condition caused by soil characteristics, and requirement of big antennas. As a strong alternative of EM, various magnetic induction (MI) techniques have been introduced. These techniques basically depend upon the magnetic induction between two coupled coils associated with transceiver sensor nodes. This paper elaborates on three basic MI communication mechanisms *i.e.* direct MI transmission, MI waveguide transmission, and 3D coil MI communication with detailed discussion of their working mechanism, advantages and limitations. The comparative analysis of these MI techniques with each other as well as with EM wave method will facilitate the users in choosing the best method to offer enhanced transmission range (upto 250 m), reduced path loss (<100 dB), channel reliability, working bandwidth (1–2 kHz), & omni-directional coverage to realize the promising MI-based wireless underground sensor network (WUSN) applications.

## INTRODUCTION

Based upon the nature of underlying medium, wireless sensor networks (WSNs) may be categorized as air-based terrestrial WSNs, soil-based underground WSNs and water-based underwater WSNs (*Akyildiz et al., 2002*). The wireless underground sensor networks

(WUSNs) (*Sardar et al., 2019*) comprise of wireless sensor devices, which operate in a subterranean environment and interact with one another wirelessly (*Banaseka et al., 2021*). These sensing nodes may be either deployed within closed underground structures like underground roads/subways, mines (*Forooshani et al., 2013*) or tunnels (*Dudley et al., 2007*) or may be buried completely under the ground. In the first scenario, in spite of sensor networks being located underground, signals are communicated through air in void space existing below earth surface which helps in improving the security in underground mines ensuring comfortable communications for the passengers and drivers in road/subway hollow tunnels as well as in securing these structures from attacks by means of consistent monitoring (*Akyildiz, Sun & Vuran, 2009*). Similarly in the second scenario, sensing nodes buried under the ground surface interact with one another through soil as propagation medium and this kind of communication of WUSNs offers a huge range of naive applications including smart irrigation (*Dong, Vuran & Irmak, 2013*), precision agriculture (*Yu, Han & Zhang, 2017*), border patrolling, soil monitoring, predicting landslides (*Aleotti & Chowdhury, 1999*) or earthquakes or volcano eruptions (*Werner-Allen et al., 2006*) and many more applications related to the Internet of Underground Things (IoUT) (*Akyildiz & Stuntebeck, 2006*; *Banaseka et al., 2021*).

The researchers observed that using the terrestrial motes in underground communication could not yield satisfactory and reliable communication results due to harsh environment of underground media (*Stuntebeck, Pompili & Melodia, 2006*). They further worked a lot on the analysis of characteristics of communication channel of WUSNs. The underground signal communication using EM wave propagation has been analysed using channel characterization models in *Akyildiz & Stuntebeck (2006)*, *Li, Vuran & Akyildiz (2007)*, *Vuran & Silva (2010)*, *Peplinski, Ulaby & Dobson (1995)*, *Silva & Vuran (2009)*, *Vuran & Akyildiz (2010)*. Here it was evaluated that how path loss and bit error rate are affected by environmental factors like water particles in soil (humidity) as well as network related factors including operating frequency and burial depth of the sensing nodes. The researchers (*Silva et al., 2015*) investigated through experiments the communication link characteristics of different types of WUSN channels such as underground to underground (UG-UG) channel, underground to above-ground (UG-AG) channel and above-ground to underground (AG-UG) channel.

However, gradually the researchers found that EM wave communication mechanism suffers from huge problems. Firstly, the high signal attenuation or path loss is caused by absorption by soil, rock elements and underground water (*Forooshani et al., 2013*). Secondly, As the soil properties such as soil type (*Salam & Vuran, 2016*) and volumetric water content, *etc.*, change very randomly with location and over time, therefore performance of sensor networks remains unpredictable (*Trang, Dung & Hwang, 2018*). Third problem is of using large sized antennas, which are deployed so that practical communication range may be achieved with low operating frequencies (*Salam & Raza, 2020*). The impact of antenna size and orientation as well as soil moisture on underground communication was also realized experimentally by developing outdoor WUSN in *Silva & Vuran (2010)*. All these issues made EM wave approach unsuitable for deploying WUSNs despite the promising application domains (*Sun & Akyildiz, 2009*).

Last decade has witnessed a new approach called magnetic induction (MI) as an effective alternative of EM communication for harsh environment like underground (*Sun & Akyildiz, 2010b*) or underwater (*Akyildiz, Wang & Sun, 2015*; *Muzzammil et al., 2020*; *Debnath, 2021*; *Domingo, 2012*). The MI channel capacity from a pair of transceiver coils has also been elaborated (*Kisseleff, Akyildiz & Gerstacker, 2014b*). The algorithms explaining the efficient mechanism of deploying the magnetic coils for WUSNs have been worked on *Sun & Akyildiz (2013)*. The authors (*Sun & Akyildiz, 2010b*, *2009*) have detailed out the analysis of path loss and bandwidth using MI communication approach in soil as underground medium. The multi-hop relay techniques are proposed to extend communication range in near-field MI communication systems (*Masihpour, Franklin & Abolhasan, 2013*). Further, gradual analysis of communication channel model has led to evolution of MI waveguide mechanism for minimizing the path loss observed in conventional EM wave approach or original MI mechanism (*Sun & Akyildiz, 2010b*). The system performance in term of path loss caused by EM wave mechanism, ordinary MI mechanism and MI waveguide mechanism have been analysed, quantified and compared with one another (*Sun & Akyildiz, 2010b*, *2009*). The reduction in path loss and increase of transmission distance (by more than 20 times) has been observed in MI waveguide communication as compared to ordinary communication by the researchers (*Liu, Fu & Wang, 2021*). Further 3D MI Coils were found to be very beneficial to be used for omnidirectional coverage (*Tan, Sun & Akyildiz, 2015*). As compared to single-dimension coils, the improvement in directionality of MI communication using multi-dimensional coils, metamaterial enhanced antennas and spherical coil-array enclosed loop antennas and polyhedral geometry has been analysed for MI-based underwater networks by researchers (*Muzzammil et al., 2020*). The researchers also provided the multimode model for characterizing the wireless channels for WSNs used in underground structures like mines for both EM and MI techniques in their study (*Akyildiz, Sun & Vuran, 2009*). The scientists also detailed out the architecture and operating framework of monitoring underground pipelines for detecting real time leakage effectively using MI-based WSN, called as MISE-PIPE (*Sun et al., 2011*). The researchers (*Sun & Akyildiz, 2010a*) have further suggested two algorithms, *i.e.* the MST algorithm and triangle centroid (TC) for effective deployment of MI waveguides to connect the underground sensors in WUSN environment. Further cross layered protocol architecture for MI-WUSNs has been explained in *Lin et al. (2015)*. A channel model considering the fading effect of presence of large obstacles on coverage performance has also been proposed and analysed for MI-based underwater wireless sensor network (*Debnath, 2021*). Further work has been done on relay or waveguide based MI-WUSNs to attain the required Quality of Service (QoS) metrics in the form of the Min-Max problem using relay selection algorithms, relay placement approaches and optimization of operational parameters (*Ishtiaq & Hwang, 2020*). In addition to EM and MI communication, the researchers have also worked on Acoustic based propagation as communication methodology, but the acoustic approach has proved to be more appropriate for detection based applications as compared to communication based applications (*Banaseka et al., 2021*). The recent advancements made

in related areas of MI-WUSNs, which might have impact on implementation of MI-WUSNs have been discussed (*Kisseleff, Akyildiz & Gerstacker, 2018*).

The aim of this paper is comprehensive elaboration of all MI techniques at one place and their comparative analysis with one another as well as with conventional EM wave approach based on thorough literature review. The various MI techniques exhibit performance enhancements with respect to different parameters. This comparative analysis will facilitate selection of appropriate technique with consideration of required parameters as per application and thereby help in finding optimized solution of application specific WUSN implementation. The paper also discusses the challenges faced by MI-based WUSN solutions as well as future scope of work based upon further exploration of MI techniques.

The intended users of this review are the researchers working on achieving QoS parameters for WUSNs using MI approach instead of EM wave methodology and thereby realizing the potential MI-WUSN applications. The researchers working in the domain of underwater wireless sensor networks may also be benefited using this analysis.

The rest of the paper is as follows. The "Methodology" section explains the methodology adopted for literature review. The "Application domains of MI-based WUSNs" section of the paper elaborates various MI-WUSN based application domains. The next section discusses the edge of MI over conventional EM wave communication. The "Various MI techniques used in WUSNs" section highlights the detailed architecture of all MI communication techniques, *i.e.* direct MI communication, MI waveguide communication and 3D MI coil communication. The next section then highlights the comparative analysis of these MI techniques. Future scope of work and conclusion is given in the last section.

## METHODOLOGY

The purpose of this paper is the detailed elaboration of all MI-based techniques followed by their comparison with one another as well as with classical EM wave approach for WUSNs. We started with a specific set of search terms used against a meta-search engine "Google Scholar" to search across multiple databases. These search words were "WUSN", "wireless underground sensor networks", "electromagnetic wave", "magnetic induction" and "MI waveguide". The next step of this systematic literature review was to gather all retrieved documents from the year 2006 to the year 2020, where the process of screening included downloading the papers published in various journals and reading their abstracts. From these, we identified all MI techniques used in WUSNs. For each of these MI techniques, further papers were searched and retrieved based on which, advantages and limitations of these MI techniques have been listed out and compared further. The literature review includes study of more than 100 research papers.

## APPLICATION DOMAINS OF MI BASED WUSNS

Unlike EM wave based WUSNs used for the cases of low burial depth utilizing UG-AG and AG-UG channels, MI-based WUSNs are especially helpful for the applications working on pure UG-UG channels, where underground sensor devices are deeply

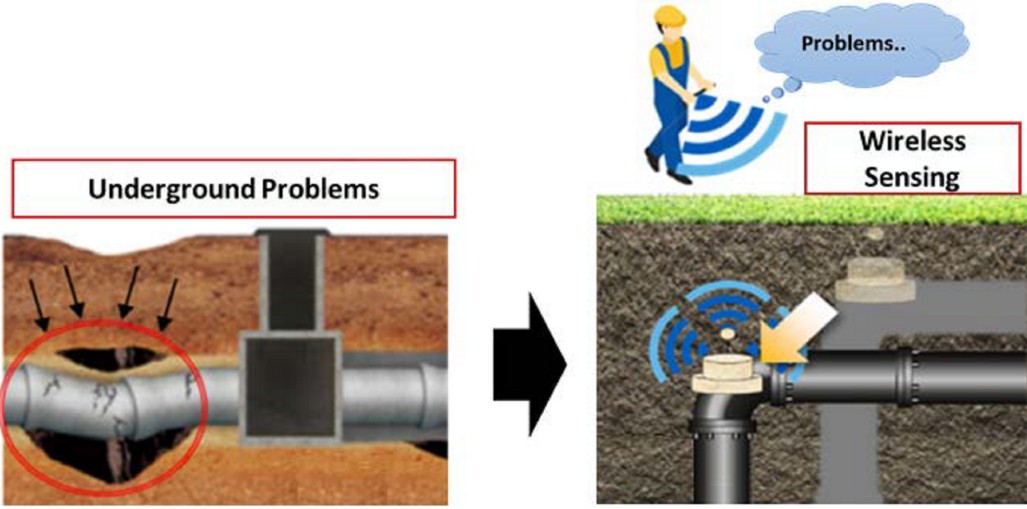

**Figure 1 Usage of MI-WUSNs for detection of leakage of water or oil (*Sun & Akyildiz, 2012a*).**

deployed in the soil or no above ground devices are there. Some of such application areas identified by the researchers are given here :

## Underground leakage monitoring applications

MI-based WUSNs may be applied for monitoring and detecting the leakage in underground infrastructures like water, gas or oil pipelines. It helps in assuring that there are no leakages in underground fuel tanks and also in determining the actual amount of oil currently available in the fuel tank so that overflowing may be avoided. MI-WUSNs also play an important role in monitoring the leakage in underground septic sewage tanks. These are used with the help of sensors deployed along the path of pipelines so as to localize and repair the leakage of gas or water from gas pipelines or water pipelines respectively (*Sadeghioon et al., 2018*) (see Fig. 1).

## Disaster prediction applications

MI-WUSNs are also used in assessing and predicting the disastrous situations like floods, eruption of volcanoes and earthquake, Tsunami, oil spilling or land sliding situations. These disastrous conditions do arise attributing to alterations in materials like soil, water *etc.* and these changes are monitored using underground sensor devices. MI-WUSNs prove far better as compared to current methods of landslide prediction, which are costlier as well as more time consuming to get deployed (*Sheth et al., 2005*). Similarly, MI-WUSNs measure and monitor the imbalances of glacier movements and volcanic eruption movements as well. All these things help in fine predictions of forthcoming natural disasters.

## Agricultural monitoring applications

MI-based WUSNs can be well utilized for monitoring applications related to agriculture, mines, tunnels, pollution and many more. The soil sensors buried inside the ground as part

of WUSN may be used for observing the soil properties like soil makeup (*Raut & Ghare, 2020*), water content in soil (humidity), density of soil *etc.* facilitating smart and efficient irrigation system (*Sambo et al., 2020*). MI-WUSNs are also beneficial for continual monitoring of methane or carbon monoxide gases inside mines which can explode and cause a big fire, if not monitored properly. These sensors are also utilized for habitat (marine life, fish farms) monitoring, exploration (natural resource) monitoring or observance of underwater pollution also.

## Underground structural monitoring applications

MI-based WUSNs are also used for monitoring the internal structures of dams, buildings to know the factors which influence the durability of these structures (*Park et al., 2005*). It is ensured by tracking stress and strain present in the material; used for their construction like water, sand, concrete *etc.*

## Sports field monitoring applications

One important application domain making use of MI-WUSNs is sports field monitoring where sensor nodes buried inside are used for monitoring the condition of soil of different types of playgrounds or sport fields like golf courses, soccer fields, baseball fields or grass tennis courts. The poor turf conditions do cause uncomfortable playing experience for the players, making it necessary to monitor and maintain the health of grass (*Akyildiz & Stuntebeck, 2006*; *Li, Vuran & Akyildiz, 2007*).

## Security related applications

MI-WUSNs are better suited for security related applications because these do have higher degree of concealment in comparison with terrestrial sensor devices. It is so because their presence is hidden and the chances of determining their presence and deactivating them are very less. By deploying pressure sensors under MI-WUSN along the border area, the concerned authorities can be alerted as soon as some illegal intruder tries to cross that region. The applications like surveillance of submarines or mines (*Rolader, Rogers & Batteh, 2004*) are also very beneficial using MI-WUSNs (see Fig. 2).

## MI-assisted wireless powered underground sensor networks

WUSNs allow for remote monitoring and management of a variety of subsurface environments, however those have a substantial reliability issue. To solve this issue and alleviate current networking issues (*Liu, 2021*) presents the magnetic induction (MI)-assisted wireless powered underground sensor network (MI-WPUSN), a new idea that combines the benefits of MI communication techniques with those of wireless power transfer mechanisms. MI-WPUSN is one-of-a-kind platform with seven envisioned devices and four various communication modes that has considerable reliability potential but is limited by its complex and difficult data collection.

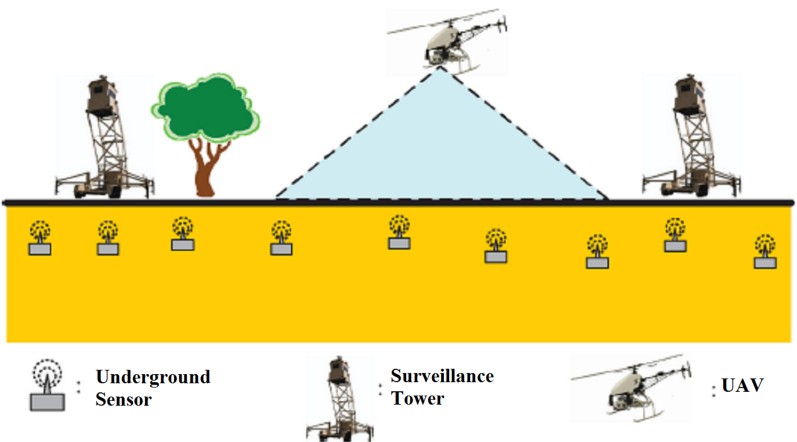

**Figure 2** Usage of MI-WUSNs for border security applications (*Vuran & Akyildiz, 2010*).

# MI AS BETTER ALTERNATIVE OF EM WAVE COMMUNICATION

The conventional signal communication techniques based upon EM wave propagation is not very much encouraging (*Huang et al., 2020*) for most of the underground communication applications due to the following reasons:

## High signal attenuation

The path loss for underground WSNs is highly dependent on a big number of parameters attributing to large variety of its underlying media such as presence of soil or rock or some fluid under the earth surface, various types of soil makeup like sand, clay or silt, volumetric water content or humidity in soil, soil density. Moreover, path loss of EM wave mechanuism is a function whereas path loss for MI-based communication is a logarithmic function of the distance (*Banaseka et al., 2021*). Due to all these parameters, signal attenuation is very high (*Akyildiz, Sun & Vuran, 2009*).

## Rapidly changing channel conditions

All the soil characteristics mentioned above vary very rapidly and unpredictably with location (like sandy soil in desert area) or time (like more water content in soil during rainfall), due to which communication channel becomes very unpredictable and unreliable. Consequently, the bit error rate (BER) also changes randomly. Due to all these, both satisfactory connectivity and energy efficiency become infeasible to be attained for WUSNs (*Trang, Dung & Hwang, 2018*; *Akyildiz, Sun & Vuran, 2009*). Even for WUSNs using Ultra Wideband (UWB) frequency of 3.1 to 5 GHz, burial depth and soil moisture were recommended to be lower than 30 cm and 20% respectively to attain acceptable signal strength (*Zemmour, Baudoin & Diet, 2016*).

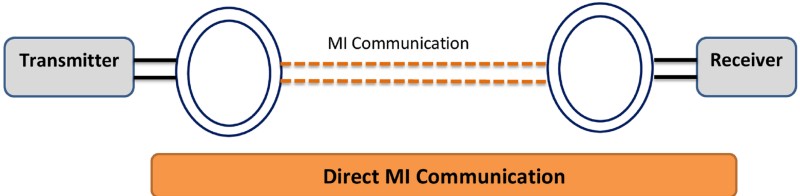

**Figure 3 Basic structure of direct MI communication (*Tan, Sun & Akyildiz, 2015*).**

## Large antenna size

For achieving communication range usable for practical applications, it becomes necessary to operate the transceivers at lower frequencies in MHz, for which it further becomes to keep the size of antenna high (*Salam & Raza, 2020*). Consequently this size of antenna becomes too large to be buried in soil (*Akyildiz, Sun & Vuran, 2009*). Even in the case of Surface Penetrating Radar used to search the underground objects, transmitting and receiving antennas were required to be moved at constant speed in linear direction to get cross-sectional image of object (*Daniels, 1996*).

To address the prominent problems of EM based WUSNs including large-sized antenna or electrical dipoles (*Tan, Sun & Akyildiz, 2015*), dynamically changing communication channel (*Akyildiz, Sun & Vuran, 2009*) and high path loss, MI technology has been identified as a far better alternative by researchers. Unlike communicating through waves, MI-based transmission mechanism makes use of near field of coil associated with transceiver sensor node (*Sun et al., 2013*). Due to the usage of small coil of wire for transmitting and receiving signal, no lower limit of coil size is required in MI communication (*Akyildiz, Sun & Vuran, 2009*).

## VARIOUS MI TECHNIQUES USED IN WUSNS

Following subsections cover in detail various MI communication methodologies—direct or ordinary MI communication, MI waveguide communication and MI three-directional communication.

### Direct or ordinary MI communication technique

The underlying architecture, working, advantages and limitations of direct or ordinary MI communication technique are detailed as follows:

#### Architecture of direct or ordinary MI communication approach

In ordinary or direct MI communication architecture (*Kisseleff, Akyildiz & Gerstacker, 2014a*), the communications signals are transmitted or received with the usage of a coil of wire using the fundamental principle of mutual magnetic induction. Resembling with the analogy of transformers, transmitted coil transmits the signal in form of sinusoidal current, which further induces another similar sinusoidal current inside the receiver node and thus communication is accomplished (*Sun & Akyildiz, 2010b*). The interaction amongst transmitter and receiver coupled coils is due to mutual induction (see Fig. 3).

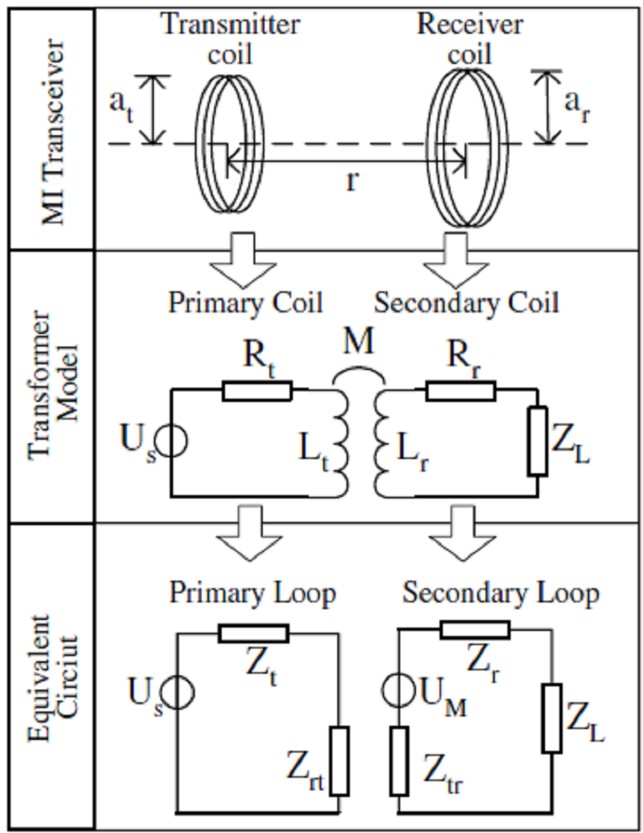

**Figure 4 Analogy of direct MI communication with a transformer (*Sun & Akyildiz, 2010b*).**

For better understanding of MI transceivers nodes, functionality of primary and secondary coil in a transformer may be referred, which is also depicted in Fig. 4. Here M indicates the mutual inductance of transmitter and receiver coils, $U_s$ is the transmitter battery voltage, $L_t$ and $L_r$ denote the self-inductance of transmitted and receiver coils. $R_t$ and $R_r$ are coil resistances and $Z_l$ denotes the load impedance of receiver node (*Tan, Sun & Akyildiz, 2015*; *Sun & Akyildiz, 2012b*; *Sun & Akyildiz, 2010b*).

Attributing to usage of near field communication, the MI coils operating at lower frequency bands can achieve more stable and reliable transmission channels in harsh WUSN medium like soil or oil (*Sun et al., 2013*).

### Advantages of direct or ordinary MI communication approach

The direct or ordinary MI transmission approach offers a promising solution to the prominent problems of EM communication viz rapidly and unpredictably changing channel conditions and need of big sized antennas.

Due to magnetic permeability of soil or rocks or water being same as that of air ($4\pi \times 10^{-7}$ H/m), MI channels are not impacted by the dynamic changes of soil with time or location (*Kisseleff, Akyildiz & Gerstacker, 2014b*; *Sun et al., 2011*). Therefore, this parameter has no effect on path loss for MI solutions. It has also been clearly observed in (*Sun & Akyildiz, 2009*) that in case of water content in soil getting increased by 25%, there

is no impact on path loss for MI solution, whereas significant path loss (upto 40 dB) is there for EM wave solution. Also instead of using huge sized antennas, small magnetic coils (of radius <0.1 m) are used for MI communication, which makes WUSNs implementable in a practical manner (*Akyildiz, Sun & Vuran, 2009*). For transmission distance of less than 1 m, path loss of MI method has been observed as smaller than EM wave method (*Sun & Akyildiz, 2009*).

### Limitations of direct or ordinary MI communication approach

Inspite of benefits of direct MI communication like stable communication channel and small size of coils, some constraints put by direct MI communication put the hindrance in making it suitable alternative for underground sensor communication applications.

**(a) Small communication range:** Although factors affecting the signal attenuation due to varying soil properties don't apply in ordinary MI communication case, but, the magnetic field created by transmitter coil gets weakened by the time it reaches the receiver coil (*Sun et al., 2013*). Due to this attenuation rate of near magnetic field, communication range attained is still too small (of the order of 10 m) for practical use for applications (*Tan, Sun & Akyildiz, 2015*).

**(b) High path loss for larger transmissions:** As the path loss in ordinary MI communication case is inversely proportional to cubic value of transmission distance as compared to simple value of transmission distance ($1/r^3$ *vs.* $1/r$) (*Sun et al., 2011*). Due to this reason, MI communication is discouraged for terrestrial WSNs. For application related to underground sensor nodes, although the path loss for MI approach caused due to soil absorption is comparatively very low when compared to EM communication, but total path loss may still be higher (greater than 100 dB) for larger transmission distances (*Akyildiz, Sun & Vuran, 2009*). Although for operating frequency exceeding 900 MHz, path loss has been observed decreasing as compared to EM wave communication (*Sun & Akyildiz, 2009*).

**(c) Performance affected due to intersection angles of two coils:** The system performance of direct MI mechanism is maximum if the transceiver nodes are deployed face to face along the same axis or in same line. But practically the intersection angle of two coils is non-zero and it significantly affects the communication performance (*Tan, Sun & Akyildiz, 2015*).

**(d) Insufficient bandwidth:** Also the bandwidth attained using ordinary MI communication approach is very very small (1–2 kHz), which is insufficient for practical applications (*Sharma et al., 2016*), whereas it is in MHz for EM wave solution. For enhancing the channel gain for direct MI communication, either the coil size may be increased or number of turns in coils may be increased. However, it results in increased size of transceivers. One more parameter influencing the channel gain is unit length resistance of the loop. Therefore lesser resistance wires and circuits may be used for ensuring lesser path loss with no increase in size. Further, it is possible to decrease the wire or circuit resistance by using better conductivity wires, better capacitors, better connectors and customized printed circuit boards (PCB) (*Tan, Sun & Akyildiz, 2015*).

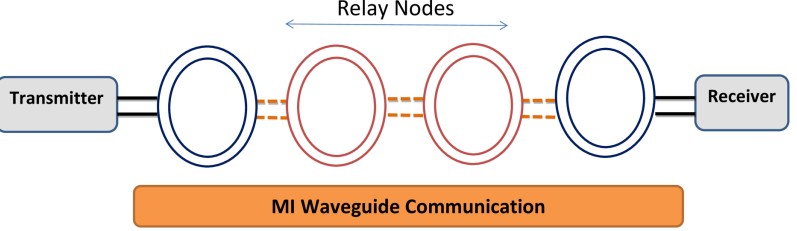

**Figure 5** Basic structure of MI waveguide technique (*Tan, Sun & Akyildiz, 2015*).

## MI waveguide communication technique

To overcome the constraints like limited communication range and high path loss observed by researchers in using direct MI mechanism, the development of an advanced technique called waveguide-based MI communication has been explored (*Sun et al., 2011*; *Sun & Akyildiz, 2010a*). This technique has proved efficient in significantly minimizing the transmission path loss, increasing the communication range and attaining a practical bandwidth of inter-sensor communication in various applications related to underground environment (see Fig. 5) (*Sharma et al., 2016*). Following subsections discuss in detail the architecture, benefits and limitations of MI waveguide technique.

### Architecture and working pattern of MI waveguide technique

The MI waveguide architecture comprises of series of multiple resonant relay coils (*Tam et al., 2020*) which are deployed between the underground transceiver sensor nodes and are wirelessly connected with one another in MI-based WUSNs (*Sun & Akyildiz, 2012a*; *Li, Vuran & Akyildiz, 2007*). Although the MI waveguide structure was initially discussed (*Tam et al., 2020*) where the relay coils used to be very near to one another resulting in strong coupling, but for MI wireless communication, the coupling between relay nodes is quite weak due to their not being very close to one another (*Sun & Akyildiz, 2010a*).

Wave technique is also used for EM communication, but unlike the relay points using EM wave technique, the MI relay point is simply a coil having no source of energy or processing device (*Sun et al., 2011*; *Sun & Akyildiz, 2012b*). The MI waveguide and the regular waveguide are based on different principles and usable for different types of applications (*Tan, Sun & Akyildiz, 2015*).

The basic working principle behind inter sensor communication in MI waveguide mechanism is the serial magnetic induction or coupling between relay nodes located next to one another (*Sun & Akyildiz, 2012a*). Although some relay nodes do exist in between transmitter and receiver devices, but even then this MI waveguide communication pattern comes under category of wireless communication. Attributing to this unique physical architecture of MI wave guide, high degree of freedom is there in deploying the nodes and utilizing them in numerous harsh conditions of underground medium (*Tan, Sun & Akyildiz, 2015*). As the MI transceiver nodes and relay coils are magnetically coupled by virtue of their placement in straight line, the relay coils will get the induced

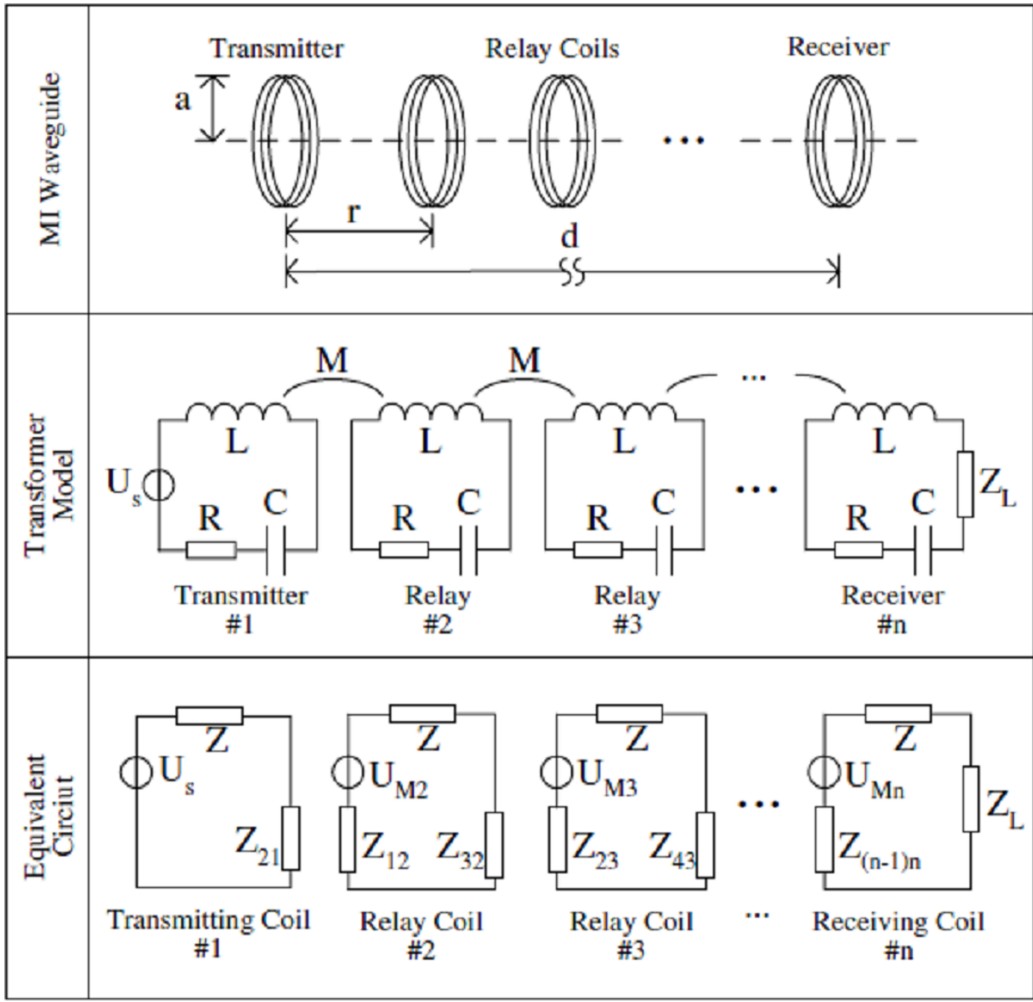

**Figure 6 Analogy of MI waveguide technique with transformer (*Sun & Akyildiz, 2010b*).**

current sequentially until it arrives at receiver node. During this entire process, signal gets strengthened till it reaches the receiver (*Tan, Sun & Akyildiz, 2015*).

In MI waveguide functionality, the sinusoidal current in the transmitter coil produces a magnetic field which varies over time, and can cause another sinusoidal current in the first relay coil and then this relay repeats the process for next relay coil and so on (see Fig. 6) (*Sun & Akyildiz, 2010b*). This allows the magnetic induction passively to be transmitted through all the intermediate coils until the MI receiver is reached, creating the MI waveguide (*Sun et al., 2011; Akyildiz, Sun & Vuran, 2009*).

It has been established that it is required to have six or more relay coils to attain effective increment of signal strength with transmission range of 2 m (*Sun & Akyildiz, 2012a*). Further relay density or number of relays can be lowered using the coils with low unit resistance and high conductivity wires and circuits (*Tan, Sun & Akyildiz, 2015*). It has been verified through physical experiments also that communication range got extended using MI waveguide system as compared to direct MI communication. Other factors like

scale, resistance value of magnetic coils and number of turns also influence the MI transmission (*Vidhya & Danvarsha, 2021*).

Figure 6 depicts MI waveguide mechanism, where n number of total coils are there, out of which $n - 2$ are relay coils which are placed equidistantly in a straight line between transceiver nodes; $r$ is the distance between the adjacent coils and d is the total communication range between transmitter and receiver node, which is same as $d = (n - 1)/r$, a is the radius of all coils, C is the capacitor with which each relay and transceiver coil is loaded. The relay coils can be made resonant coils for effective transmission of magnetic signals by appropriate design of capacitor value. Between any two adjacent relay coils, mutual inductance does exist, whose value is based on their inter-coil distance (*Sun & Akyildiz, 2012a*).

### Advantages of MI waveguide approach

By using MI waveguide approach for underground communication, following benefits have been observed:

**(a) Stable MI channel:** As most of the underground transmission media *i.e.* soil is non-magnetic having almost equal permeability values, therefore rate of attenuation of magnetic fields created by coils remains almost unaffected, keeping the MI channel conditions constant and stable (*Sun & Akyildiz, 2013*, *2012a*).

**(b) Need of a smaller number of sensors:** In case of underground communication using MI waveguide, distance of 5 m is kept between two relay nodes, which is even more than the maximum transmission range attained during EM wave communication (*Sun & Akyildiz, 2012a*). As shown in Fig. 6, instead of using $n$ number of full fledged transceiver sensor nodes $(n - 2)$ relay nodes are used with two transceiver nodes on transmission and receiving sides, which clearly points to the requirement of less number of full fledged underground transceiver nodes (*Sun et al., 2011*; *Sun & Akyildiz, 2010b*).

**(c) Bandwidth:** The bandwidth achieved for both ordinary MI communication as well as MI waveguide communication is in the same range (1–2 kHz), which has been found sufficient for non-traditional media applications requiring low data rate monitoring (*Sharma et al., 2016*; *Kisseleff, Akyildiz & Gerstacker, 2014a*). Also, if operating frequency is 10 MHz, then 3-dB bandwidth of MI waveguide is found to be in same range as of direct MI communication *i.e.* (1–2 kHz) (*Sun & Akyildiz, 2010b*).

**(d) Path loss reduction:** Due to placement of relay coils between transceiver sensor nodes, MI waveguide mechanism offers huge reduction of path loss, which is the most prominent advantage of this technique (*Dudley et al., 2007*). More specifically, this is attributed to appropriate design of waveguide parameters (*Sun et al., 2011*; *Akyildiz, Sun & Vuran, 2009*). The analysis in *Sun & Akyildiz (2010b)* has shown that MI waveguide offers path loss smaller than 100 dB for distance even more than 250 m, whereas for transmission range of even slightly more than 5 m, same path loss of 100 dB is observed for EM wave system as well as for direct MI communication. By reducing the relay distance and resistance value of coil wire, path loss can be further lessened for MI waveguide (*Liu, Fu & Wang, 2021*).

**(e) Extension of transmission range:** Using MI waveguide technique, the transmission range is significantly extended as compared to EM wave communication or ordinary MI communication (*Sharma et al., 2016*; *Sun & Akyildiz, 2013*). It has also been established experimentally that if Mica2 sensor are used for underground communication in soil using EM wave mechanism, communication range achieved is less than 4 m, which increases to 10 m with similar device size and power for direct MI communication and further gets extended to more than 100 m for MI waveguide communication (*Sun et al., 2011*; *Sun & Akyildiz, 2010a*). The distance between relay nodes is even more than the maximum communication range of EM wave transmission (*Sharma et al., 2016*). Due to this extension in transmission range using MI waveguide mechanism, a fully connected sensor network may be attained even without deploying large number of sensors (*Sun & Akyildiz, 2012a*).

It has also been observed by the researchers that with the increase in transmission distance, the transmission power gets decreased (upto 50% of power required for EM or direct MI) making it favourable for the energy-constrained non-traditional media (*Sharma et al., 2016*).

**(f) Better robustness and easier deployability and maintenance:** Unlike a real waveguide, the MI waveguide is not a continual structure and therefore is comparatively more flexible and easy to be deployed and maintained at every 6 to 12 m (*Sun & Akyildiz, 2012a*; *Akyildiz, Sun & Vuran, 2009*). The relay coils in MI waveguide don't need extra power due to passive relaying of magnetic induction (*Sun & Akyildiz, 2010a*). Hence unlike the sensor devices, these relay coils are easily deployable and once buried in soil, don't need much more regular maintenance (*Sun & Akyildiz, 2013*). If due to some harsh conditions, some of the relay coils get damaged, even then the remaining relay coils ensure robustness of sensor network (*Sun et al., 2011*).

**(g) Cost:** As the relay coils used in MI waveguide consume no energy and unit cost of these relay coils is very less (*Sharma et al., 2016*; *Sun & Akyildiz, 2012a*; *Akyildiz, Sun & Vuran, 2009*), therefore overall cost of underground sensor network gets reduced to large extent as compared to using expensive relay sensor devices in EM wave communication (*Sun & Akyildiz, 2012b*).

**(h) Prolonged system lifetime:** MI waveguide technique also leads to prolonged lifetime of system because the underground sensor devices equipped with MI transceiver nodes can be recharged by above ground charging devices using inductive charging mechanism (*Sun & Akyildiz, 2013*; *Sun & Akyildiz, 2012a*; *Sun & Akyildiz, 2010b*). In RF-challenged environments, it becomes very cumbersome to replace device batteries; therefore this option of magnetic induction charging proves very beneficial (*Sun & Akyildiz, 2010a*).

### Limitations of MI waveguide approach

Inspite of numerous advantages offered by MI waveguide technique for WUSNs, some constraints have been observed by the researchers, which highlight the essence of more optimization work on MI waveguide approach.

**(a) Limited channel capacity and data rate:** It has been found that due to lower ratio (order of 2.5) of mutual induction to self-induction (also termed as relative magnetic coupling strength) between adjacent relay coils working at the resonant frequency to attain low path loss in MI waveguide approach, the channel bandwidth becomes very limited (1–2 kHz) (*Sun & Akyildiz, 2010a*). This decrease in channel bandwidth becomes more adverse, if communication distance increases to a particular threshold value, which finally leads to lower channel capacity as well as unsatisfactory data rate, inspite of large communication range (*Sun & Akyildiz, 2010a*).

**(b) Usable for limited application domains:** Attributing to data rate and bandwidth limited channels, MI waveguide technique may be adopted for those applications only, where required data rate is low (*Sun & Akyildiz, 2012a*). For WUSN applications like rescuing the trapped ones in underground mines or border patrolling, large amount of data is required to be timely transmitted on MI channels, which requires higher data rate and bandwidth. Hence much efforts are needed for enhancing the MI channel capacity for MI-WUSNs (*Sun & Akyildiz, 2010a*).

**(c) Reliability issue:** As the multiple resonant MI relay coils constitute the foundation of communication success of MI waveguide approach, hence overall performance of such sensor networks are based not only on the transceiver sensor nodes, but also these relay coils. Therefore, the issue of reliability of such underground sensor networks is needed to be analysed in tough underground media (*Sun & Akyildiz, 2012a*).

**(d) Complex deployment mechanism:** Due to very rough and hostile underground communication medium, all transceiver sensor nodes are isolated until and unless connected by MI waveguide mechanism. Therefore, deployment of large number of relay coils in MI-WUSNs costs a big amount of labor (*Banaseka et al., 2021*) and therefore needs very thoughtful and complex strategies (*Sun & Akyildiz, 2013*) targeting at the objective of building a connected robust wireless sensor network with minimal possible relay coils (*Sun & Akyildiz, 2013*).

It was established by the researchers that for underground pipelines made up of metallic material, no or very few relay coils are required due to metal pipe itself working as magnetic core of MI waveguide. For non-metallic pipes such as PVC, single relay coil deployed around 5 m distant from one another is enough. For winding these relay coils, the underground pipeline proves to be perfect core leading to small coil deployment cost if they are winded on pipeline during deployment time itself (*Sun et al., 2011*).

**(e) Lack of omnidirectional propagation:** Most of the channel characterizations have been done with assumption of placement of transceivers or relay coils in straight line, which is practically not always true. For transceiver nodes based on MI communication, the strength of received signal at receiver end is affected by the angle between axes of two mutually coupled coils. To maintain high-quality transmission in such cases, multidimensional MI coils are developed and deployed (*Tan, Sun & Akyildiz, 2015*).

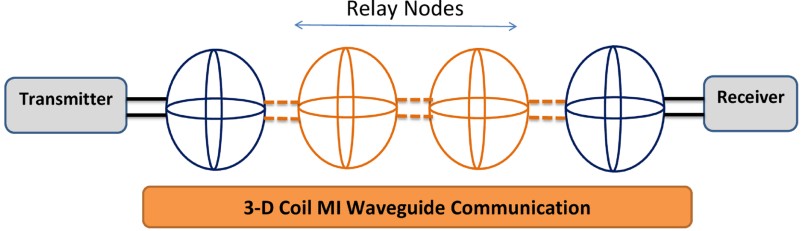

**Figure 7** Basic structure of MI waveguide with 3D coils (*Tan, Sun & Akyildiz, 2015*).

## MI 3-D coil communication technique

The basic architecture, functionality and advantages for MI 3-D coil communication technique are detailed out in following subsections:

### Architecture of 3D coil MI communication approach

In most of the practical underground communication applications, coils are not buried in straight line due to two probable causes, first being inability to deploy relay coils in exact planned positions due to rocks or pipes being already present inside the ground and secondly the positions of already buried coils may get changed during operation of network due to above ground pressure or movement of soil. It is also well established that received signal strength at MI transceiver end is affected by the angle between the axes of two adjacently placed coils. Therefore, option of using multidimensional MI coils in such a complex scenario has been worked upon for high quality transmission between sensing nodes (*Tan, Sun & Akyildiz, 2015*). More precisely, 3-directional (3D) coils have been designed and used which offers omni-directional signal coverage as well as minimal number of coils leading to reduced system complexity and cost (Refer Fig. 7) (*Tan, Sun & Akyildiz, 2015*).

### Advantages of 3D coil MI communication approach

In 3D MI coil system, three individually fabricated unidirectional (UD) coils are vertically mounted on a cubical structure having each side with length of 10 cm such that each 3D coil is perpendicularly deployed with respect to others. These three coils are meant for forming a powerful beam along three different axes of Cartesian coordinate (*Ishtiaq & Hwang, 2020*). As the magnetic flux created by one surrounding coil becomes zero with respect to another two orthogonal coils, hence these three coils do no interfere with each other, attributing to field distribution structure of coils. Similar to direct MI coils, these 3D MI coils are also made up of 26-AWG wire and each one of these coils is supported by a serial capacitor for detecting resonance. At the side of receiving node, all the three signals from three coils are added (*Liu, Fu & Wang, 2021*).

Depending on channel model, after fixing MI coil parameters and transmission distance, it is the intersection angle between transmission and receiver nodes, which determines the signal strength. At least one coil can achieve adequate signal strength with three orthogonal coils regardless of how much the angle of intersection is changed. Even if MI coils are rotated or intersection angel between those gets changed, high degree of

**Table 1 Comparative analysis of EM wave, direct MI, MI waveguide and 3-D coil MI waveguide communication methods.**

| Sr. No | Parameter | EM wave communication | Direct MI communication | MI waveguide | MI waveguide with 3-D coils | References |
|---|---|---|---|---|---|---|
| 1 | Transmission range | up to 10 m | up to 10 m | up to 250 m | up to 250 m | *Sun & Akyildiz (2009), Salam & Vuran (2018)* |
| 2 | Path loss (on low operating frequency, <900 MHz) | Lower (<100 dB) (2nd order function of transmission range) | Higher (>100 dB) (6th order function of transmission range) | Lower (<100 dB even for 250 m distance) | Same as MI waveguide | *Sun & Akyildiz (2009, 2010b), Liu, Fu & Wang (2021)* |
| 3 | Path loss (on high operating frequency, > = 900 MHz) | Increases | Decreases | Decreases | Decreases | *Sun & Akyildiz (2009, 2010b), Liu, Fu & Wang (2021)* |
| 4 | Effect of increasing VWC (by 25 %) on Path Loss | Path loss Increases by 40 dB | Unaffected | Unaffected | Unaffected | *Reddy et al. (2020), Zemmour, Baudoin & Diet (2016)* |
| 5 | Bandwidth | Higher (In MHz) | Lower (1–2 kHz) | Lower (1–2 kHz) | Lower (1–2 kHz) | *Sun & Akyildiz (2010b), Reddy et al. (2020), Salam & Vuran (2018)* |
| 6 | Antenna size | Large | Small (coil of radius <0.1 m) | Same as Direct MI | Same as Direct MI | *Sun & Akyildiz (2009, 2010b), Zemmour, Baudoin & Diet (2016)* |
| 7 | Required transmission power | Higher | Higher | Lower (less than 50% of EM and Direct MI) | Same as MI waveguide | *Sun & Akyildiz (2009)* |
| 8 | Channel condition | Unstable | Stable | Stable | Stable | *Reddy et al. (2020)* |
| 9 | Omni directional coverage | No | No | No | Yes | *Ishtiaq & Hwang (2020)* |

communication is supposed to be kept by the system in this case (*Tan, Sun & Akyildiz, 2015*). It has also been observed that if transmission gain is maximized using optimal power allocation and adoption of spatial-temporal code, good system performance can be achieved by combining the received signals at three orthogonally placed coils (*Sharma et al., 2016*).

In underwater sensor networks also, it has been found through modelling and analysis that the transmission range of MI system achieved is of the order of 20 m range using small sized coils of 5 cm radius with high value of water conductivity. Therefore, using 3D coils helps in establishing more robust MI links, which remain unaffected by dynamical rotation of sensor nodes (*Guo, Sun & Wang, 2015*).

## COMPARATIVE ANALYSIS OF VARIOUS MI COMMUNICATION METHODS

After detailed study of all physical layer techniques, it is clear that for applications having sensor nodes deeply buried inside soil, MI is a better technology as compared to EM wave technology. All three MI transmission techniques have their relative advantages and limitations.

The comparative analysis of EM wave technique as well as all MI techniques has been summarized in Table 1. As compared to EM wave communication, direct MI

communication is seen as a better option, if factors like dynamic channel conditions of underlying media *i.e.* soil, need of large antenna size or effect of volumetric water content (VWC) percentage on path loss and connectivity are taken into consideration. EM wave approach offers better bandwidth as compared to low bandwidth of 1–2 kHz achieved using MI approach. The comparison of path loss parameter is quiet complex, as its behaviour is different in different scenarios. For very near region (transmission distance, $d < 1$ m), Direct MI method exhibits smaller path loss with respect to EM technique, but beyond this, path loss for MI channel becomes even 20 dB more than that of EM technique. Also, for dry soil, EM path loss is lesser, but with the increase of VWC in soil (which is generally the case), path loss of EM wave keeps on increasing, making MI as better alternative for such cases (*Sun & Akyildiz, 2009*). Also, as path loss is inversely proportional to operating frequency in EM method and directly proportional for MI method, therefore for operating frequencies greater than 900 MHz, path loss decreases for direct MI approach (*Sun & Akyildiz, 2009*). The MI waveguide approach is better than both EM wave as well as direct MI approach. The communication range offered by both EM wave and direct MI is not enough for practical applications. Here, MI waveguide approach proves better which offers communication distances almost 25 times that of direct MI or EM communication (*Sun & Akyildiz, 2010b*). The MI waveguide transceivers require only less than half of energy consumed by EM wave or MI method, making MI waveguide suitable for energy constrained applications in addition to lower overall cost due to relay nodes not requiring any power (*Sun & Akyildiz, 2009*). For both ordinary MI as well as MI waveguide system, the bandwidth achieved is smaller of the range 1–2 kHz, which is far lesser than EM wave mechanism, but it suffices for applications of low data rate monitoring (*Sharma et al., 2016*; *Reddy et al., 2020*). The MI systems (the MI waveguide as well as 3D MI coil) exhibit constant channel condition and offer relatively longer transmission range than that of the EM wave-based system. All the characterizations of path loss or bit error rate or transmission distance of direct MI or MI waveguide techniques have been done with assumption of sensors nodes deployed in straight line, which is not the case in reality. It makes MI 3-Directional coil mechanism as best option to offer omnidirectional coverage keeping other benefits same (*Sun & Akyildiz, 2010b*; *Akyildiz, Sun & Vuran, 2009*; *Ishtiaq & Hwang, 2020*).

# RESEARCH CHALLENGES AND SCOPE FOR FUTURE WORK

There are a number of challenges before utilizing MI-WUSNs which need further attention, exploration and research work. As clarified in previous section, bandwidth achieved in all MI techniques is very small, due to which WUSN applications requiring high data rate monitoring can not be implemented. The combined usage of active and passive relaying in MI waveguide offering low path loss is another challenging area due to significant design constrains attributing to determining appropriate location and operation pattern of each relay node (*Tam et al., 2020*). Although using orthogonal or 3D coils boosts the signal quality and other system parameters but designing such coils is also quite challenging and needs further work. One more area open for future work is

interaction of MI-WUSNs with other types of WSNs, such as WUSNs interfacing with underwater WSNs in case of exploration of deep oceans or WUSNs interacting with power grid for monitoring structural health or WUSNs communicating with self-driving cars in case of navigation and charging. The upgradation of presently available solutions, taking care of robust adjustment is also a big challenge because slight deviation in either of the system parameters may lead to rendering whole theoretical solution as invalid due to imperfect channel state information (CSI). One example of such scenarios is varying soil wetness during rainfalls which may result in additional critical modification of channel state. One of the most promising but less explored areas of future work for MI-WUSNs is to design cross-layer architecture ensuring multi-objective optimization (*Singh, Singh & Singh, 2021*) which could optimize system performance in term of throughput, charging capability and accuracy of localization. Such types of multidimensional optimization leading to design of self-charging and power-efficient networks with the constraint of high system performance based upon application or operation mode is still an open area for future work for researchers (*Kisseleff, Akyildiz & Gerstacker, 2018*). Estimating multi-hop route using the static and mobile relay nodes and establishing deterministic channel state models are also the areas requiring further research work (*Ishtiaq & Hwang, 2020*). MI communication is an effective and reliable communication mode for underground sensors to interact with each other. Meanwhile Wireless Power Transfer (WPT) mechanism is used to remotely charge the sensor batteries in WUSNs ensuring better network reliability. Integration of MI and WPT has large potential and scope for improvement of reliability and practicability of WUSNs, especially in situations when no human support is possible (*Liu, 2021*).

## CONCLUSION

Using wireless sensor networks in non-conventional media such as soil has paved the way to a large number of novel applications ranging from soil monitoring and underground infrastructural monitoring to border security related applications. The transmission constraints such as dynamic channel condition, high path loss and large antenna size put by EM wave communication mechanism for WUSNs have been addressed using MI communication. This technique is based on the basic principle of mutual induction between coils connected with transceiver nodes of WUSNs. This work has detailed out the gradual progression from ordinary MI communication to MI waveguide technique to MI waveguide with 3-D coils. These MI techniques have proved fruitful to offer advantages like constant channel condition due to similar permeability of propagation medium (air, water, rocks), reduced path loss due to low-cost and passive relay coils deployed between the transceivers, enhanced communication range attributing to relay coils, feasible bandwidth, negligible propagation delay, and small-sized coils. The comparative analysis of these MI techniques made in the work has established that MI waveguide using 3D coils is the best technique for practically realizing WUSN applications. The future scope and open challenges discussed in the work further opens various research avenues for the researchers in time to come.

### Funding

The authors received no funding for this work.

### Competing Interests

The authors declare that they have no competing interests.

### Author Contributions

- Pratap S. Malik conceived and designed the experiments, performed the experiments, analyzed the data, prepared figures and/or tables, authored or reviewed drafts of the paper, and approved the final draft.
- Mohamed Abouhawwash conceived and designed the experiments, performed the experiments, analyzed the data, prepared figures and/or tables, authored or reviewed drafts of the paper, and approved the final draft.
- Abdulwahab Almutairi conceived and designed the experiments, performed the experiments, analyzed the data, prepared figures and/or tables, authored or reviewed drafts of the paper, and approved the final draft.
- Rishi Pal Singh conceived and designed the experiments, performed the experiments, analyzed the data, prepared figures and/or tables, authored or reviewed drafts of the paper, and approved the final draft.
- Yudhvir Singh conceived and designed the experiments, performed the experiments, analyzed the data, prepared figures and/or tables, authored or reviewed drafts of the paper, and approved the final draft.

### Data Availability

This is a literature review article, no specific data or code was used.

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
