# Peer review of "Comparative analysis of magnetic induction based communication techniques for wireless underground sensor networks"

_PeerJ Computer Science, doi:10.7717/peerj-cs.789_

## Round 0.1 · original submission · Major Revisions

Date: 05/04/2021

Your manuscript has not been recommended for publication in PeerJ Computer Science in its current form; however, we do encourage you to address the concerns and criticisms of the reviewers detailed at the bottom of this letter and resubmit your article once you have updated it accordingly.

Please revise your manuscript based on reviewers’ feedback and resubmit; elaborate on your points and clarify with references, examples, data, etc. If you do not agree with the reviewers’ views, then include your arguments in the updated manuscript. Also, note that if a reviewer suggested references, you should only add ones that will make your article better and more complete.

We highly recommend that you review the grammar one more time before resubmitting.

In addition, please provide quantitative performance achievements of this work in the "Abstract".

Reviewer 1 ·

Basic reporting

There are few format and spelling problems, and also unprofessional when claim "accurate predictions" without giving data. Examples are as follows:
1. "Table 1. Comparative analysis of EM wave, Direct MI, MI waveuide and 3-D Coil MI waveguide communcation methods" misspelling of "Waveguide" and "communication".
2. Line 29, WUSN, needs to be expended as it’s first time appears.
3. Line 137, “…accurate predictions of …”, define what is the (accurate) range, resolution and sensitivity of the prediction?
4. Line 306, “the required number of underground transceiver nodes may be reduced to a significant level”, at what level?
5. Line 311, “MI waveguide mechanism offers huge reduction of path loss”, what is the loss level?

Experimental design

As the statements from the Introduction and from the Survey Methodology are inconsistent, the authors should decide "What is the objective of this paper?"
In line 90 of the introduction, the authors described that “This literature review is needed and intended to do the comparative analysis of all the techniques so as to enable the users to find the optimal solution of MI based WUSNs for respective applications”, while in line 107 of the Survey Methodology, “The purpose of this paper is to review the usage of magnetic induction approach in comparison with conventional EM wave approach for WUSNs and subsequently review all available MI techniques used for the same. ”
The authors should decide whether to compare the techniques among MI approach or compare the MI with the conventional EM wave approach?

Validity of the findings

1. This paper is a literature review of magnetic induction mechanism for WUSNs, however the authors should point out what is the novelty of this paper?
2. The authors summarized the comparision of EM wave, Direct MI, MI waveguide and 3-D Coil MI waveguide communication methods in Table 1. However, clear numbers or range for parameters should be provided instead of "Higher/ High/ Lower/ Low" as in table 1.

Additional comments

This paper is a literature review of magnetic induction mechanism for WUSNs, the authors compared the three techniques in magnetic induction approach, and with the classical electromagnetic wave.
However, the authors can further improve the manuscript in several aspects:
1. Your introduction needs to support the objective. At the beginning of the introduction, the authors detail discussed the classical electromagnetic wave approaches (3 paragraphs), and their disadvantages. The authors should decide what is the main objective of this paper? (Comparative analysis of magnetic induction? as in title) or (comparison of MI with classical electromagnetic wave as described in Methodology?) The introduction should work for introducing the main objective.
2. Conclusions are lack of data support. e.g. line 228-243, 301-343
3. What are the main challenges in magnetic induction based communication?
4. Mixed of capital and lower case in sentences. e.g. in title "Comparative Analysis of Magnetic induction based Communication for wireless underground sensor networks", line 19 "......Electromagnetic wave (EM)......", line 23 "......Electromagnetic wave (EM)......" line 188,......
5. line 166. indentation of f)......

Reviewer 2 ·

Basic reporting

1. In line 474 and 476, two references are same.
2. Figure 7 should show the 3-D structure more clearly.
3. Literature research is not sufficient.

Experimental design

As a literature review paper, most references are too early, more latest should be listed. For example, from line 397 to 417, the reference of advanced 3D-MI technique are published in 2016.

Validity of the findings

Lack of quantitative analysis for each technology. For example, more quantitative data should be presented in Table 1. More figures of analysis or comparasion of the results shoud be presented.

Additional comments

Authors should strengthen literature review in related fields. There should be more academic content, such as proof of the novelty of the method and quantitative analysis, rather than vague written statements.

---

## Round 0.2 · Major Revisions

After the first round of revision, reviewers felt that the work is required a major revision based on the following reasons. Hence, we do encourage you to address the concerns and criticisms of the reviewers detailed at the bottom of this letter and resubmit your article once you have updated it accordingly.

1. As this is a review paper, the given reference list doesn’t capture all the relevant references from all published works from different research group in different parts of the world. It is clear that majority of the references are from the research group from I.F. Akyildiz and Z.Sun etc. Please carefully review this to diversify the references.

2. In many places of the paper, reference was not given or missing. Authors should address this comment carefully. For example: (a) under “SURVEY METHODOLOGY: APPLICATION DOMAINS OF MI BASED WUSNS: a) Underground leakage monitoring applications”, (b) under “SURVEY METHODOLOGY: APPLICATION DOMAINS OF MI BASED WUSNS: e) Sports field monitoring applications”(c)
3. Under “MAGNETIC INDUCTION AS BETTER ALTERNATIVE OF EM WAVE COMMUNICATION”, some RF/Microwave/Sensors/Antennas references should be given here. If authors only used the references from other paper which is a review paper, this review paper will not add any value to the published domain as this information is available from other published works. Insightful review, discussions and citations are required here to enhance the quality of this paper.

4. This comment is related to the previous comment. In section “COMPARATIVE ANALYSIS OF THREE MI COMMUNICATION METHODS”, Table 1 shows the comparison between EM wave communication, Direct MI, MI waveguide and 3D coil MI waveguide. I cannot see any reference or detailed discussions, i.e. antenna size, transmission range, bandwidth, etc on the EM wave communication. Not sure how the given references were used to cover the EM wave communication as they are for MI waveguides.
Authors should consider some of the following references if related to EM wave communication topics.

‘A theoretical model of underground dipole antennas for communications in internet of underground things’, IEEE Trans. Antennas Propag., 2019, 67, (6), pp. 3996–4009

‘Link budget maximization for a mobile-band subsurface wireless sensor in challenging water utility environment’, IEEE Trans. Ind. Electron., 2018, 65, (1), pp. 616– 625

‘Improved communications in underground mines using reconfigurable antennas’, IEEE Trans. Antennas Propag., 2018, 66, (12), pp. 7505–7510

Opportunities and Challenges in Health Sensing for Extreme Industrial Environment: Perspectives From Underground Mines," in IEEE Access, vol. 7, pp. 139181-139195, 2019,

Design of mobile band subsurface antenna for drainage infrastructure monitoring, IET MAP, 2019,
"Soil effects on the underground-to-aboveground communication link in ultrawideband wireless underground sensor networks", IEEE Antennas Wireless Propag. Lett., vol. 16, pp. 218-221, 2017.

5. The clarity of all the figures of this paper should be enhanced. Authors should review this carefully.

Reviewer 2 ·

Basic reporting

1. The literature was not well modified. Only a few new literatures were added without detailed interpretation.
2. Some pictures are not clear enough. It is better to use vector pictures.

Experimental design

no comment

Validity of the findings

no comment

Additional comments

1. Please add the latest references, and they must be interpreted in detail in the text, not simply cited.
2. Please increase the clarity of the picture.
3. The references on which key technologies are described are still too old.

---

## Round 0.3 · Major Revisions

After carefully reading the comments, unfortunately, the revision is unacceptable due to authors failed to indicate how they addressed and implemented all the comments in the revised manuscript. Owing to this,
Please revise your manuscript based on last round of reviewers’ feedback and resubmit; elaborate on your points and clarify with references, examples, data, etc. If you do not agree with the reviewers’ views, then include your arguments in the updated manuscript. In addition, please clearly indicate in the responses where are the changes in the revised manuscript,

In addition, we highly recommend that you review the grammar one more time before resubmitting and the quality of the figures should be improved.

---

## Round 0.4 · Major Revisions

Date: 24/08/2021

The reviewer has expressed concerns about not providing up-to-date references and related content in this review paper.

To address this, authors should carefully review all the new literature (last three years, i.e. 2019, 2020, 2021) and provide the details of the development in the proposed research.

Reviewer 2 ·

Basic reporting

1. Authors did not follow the previous comments and the references are still very old. This is not a qualified review paper.
2. The overviews of Figure 5 and Figure 7 are better put together.
3. In page 14, authors only added two technologies in the last 2 rows of the table without describing them.
Above all, authors should focus on reviewing newer technologies.

Experimental design

no comment

Validity of the findings

no comment

Additional comments

no comment

---

## Round 0.5 · accepted · Accept

Authors have addressed all the comments from the reviewers. This paper is recommended to publish in its current form.

Reviewer 2 ·

Basic reporting

no comment

Experimental design

no comment

Validity of the findings

no comment

Additional comments

no comment